

# Parameter transferability of a distributed hydrological model to droughts

Giulia Bruno[1,2], Doris Duethmann[3], Francesco Avanzi[1], Lorenzo Alfieri[1], Andrea Libertino[1], and Simone Gabellani[1]

[1]CIMA Research Foundation, Via Armando Magliotto 2, 17100 Savona, Italy
[2]University of Genoa, Viale Causa 13, 16145 Genova, Italy
[3]IGB Leibniz Institute of Freshwater Ecology and Inland Fisheries, Müggelseedamm 310, 12587 Berlin, Germany

**Correspondence:** Giulia Bruno (giulia.bruno@cimafoundation.org)

**Abstract.** Hydrological models often have issues in simulating streamflow (Q) during droughts, because of hard-to-capture feedback mechanisms across precipitation deficit, actual evapotranspiration (ET), and terrestrial water storage anomalies (TWSA). To gain more insights into these performance drops and move toward more robust hydrological models in the anthropogenic era, we evaluated Q, ET, and TWSA simulations during droughts of different severity and their sensitivity to the climatic conditions of the calibration period. We used the distributed hydrological model Continuum over the heavily human-affected Po river basin (northern Italy, period 2010 - 2022) and independent ground- and remote sensing-based datasets of Q, ET, and TWSA as benchmarks. Across the 38 study sub-catchments, Continuum simulated Q comparably well during wet years (2014 and 2020) and moderate droughts (2012 and 2017) with mean KGE = 0.59±0.32 during wet years and = 0.55±0.25 during moderate droughts. The model simulated well Q for the outlet section of the basin also for the severe 2022 drought (KGE = 0.82). However, performances for 2022 declined across the other sub-catchments (mean KGE = 0.18±0.69, meaning the model still preserved some skill over a climatological mean). The model properly represented seasonality of Q, ET, and TWSA over the basin, as well as a declining trend in TWSA. We explained the performance drops in 2022 with an increased uncertainty in ET anomalies, in particular in human-affected croplands. Calibrating during a moderate drought (2017) did not improve model performances during the severe 2022 drought (mean KGE = 0.18±0.63), pointing to the fairly unique conditions of this period in terms of hydrological processes and human interference on the hydrological cycle. By highlighting increased uncertainty of hydrological models specifically during severe droughts which are expected to increase in frequency, these findings provide relevant guidelines for assessments of model robustness in a changing climate and so for informing water management, disaster risk reduction, and climate change adaptation strategies.

## 1 Introduction

Droughts affect all components of hydrological systems (Van Loon, 2015) and can have severe and multifaceted impacts (Erian et al., 2021). A warming climate may lead to an increase in drought impacts (Naumann et al., 2021). Robust modelling of water availability throughout the whole hydrological cycle during droughts is essential to inform water management, disaster risk reduction, and climate change adaptation strategies.



Distributed process-based hydrological models allow spatial estimates of hydrological fluxes and states (Fatichi et al., 2016),
even at large scales and hyper-resolutions (< 1 km, Bierkens et al. (2015)). Such models are increasingly being used for
assessments of climate change impacts on droughts (Van Huijgevoort et al., 2014; Cammalleri et al., 2020b; Dembélé et al.,
2022), drought monitoring (Cammalleri et al., 2017, 2020a; Saha et al., 2021), forecasting (Trambauer et al., 2015; Van Hateren
et al., 2019; Sutanto et al., 2020), and drought studies in general (Mastrotheodoros et al., 2020; Yang et al., 2021; Rakovec
et al., 2022). However, some studies revealed reduced model performances when simulating streamflow droughts (Kumar
et al., 2022) and their generating processes (Van Loon et al., 2012; Avanzi et al., 2020). Further, human activities can heavily
modify the hydrological cycle (Abbott et al., 2019) and streamflow droughts (Van Loon et al., 2022), but their representation
in hydrological models remains challenging (Wada et al., 2017).

More broadly, many widely used hydrological models have drops in streamflow (Q) modelling skills when simulating periods
with climatic conditions that differ from those of the calibration period (Klemeš, 1986; Li et al., 2012; Duethmann et al., 2020).
Such issues in the transferability of model parameterizations (hereinafter model transferability) can pose challenges in correctly
simulating Q during droughts (Saft et al., 2016). Some studies suggested that including dry periods in the calibration improves
Q simulation during droughts (Li et al., 2012; Yang et al., 2021), but results in this regard are still inconclusive (Avanzi et al.,
2020). Further, previous studies revealed that model deterioration in Q simulation during droughts may be related to poor
simulation of actual evapotranspiration (ET, Avanzi et al. (2020)) or Terrestrial Water Storage Anomalies (TWSA, Westra
et al. (2014); Fowler et al. (2020)). For instance, Avanzi et al. (2020) showed that a semi-distributed hydrological model had
statistically significant drops in Q and ET performances during the 2012-2016 drought over a Californian river basin, while
not in the simulation of subsurface storage; thus, they argued that the poor representation of ET, and its climate elasticity in
particular, drove the deterioration in Q modelling skills. Further, Fowler et al. (2020) found that in catchments where Q was
poorly simulated by common lumped conceptual models during the Millennium drought in south-east Australia, the models
also failed in reproducing long-term decline in storage. This highlights that evaluating hydrological models against multiple
hydrological fluxes and states may represent a way to analyse causes of poor model transferability, verify model internal
consistency and so move towards more robust modelling (Guo et al., 2017). Today ET and TWSA remote sensing-based
products can be particularly useful for model evaluation, especially for distributed models (Rakovec et al., 2016b; Hulsman
et al., 2021; Bolaños Chavarría et al., 2022) as they allow to check also their spatial representativeness. Nonetheless, assessment
of model transferability to severe droughts using independent and spatially distributed ET and TWSA remote sensing-based
products is still rare.

To contribute to fill this research gap, the questions we aimed to answer here are: (i) is the expected drop in performance
of distributed hydrological modelling for Q simulation sensitive to drought severity?; (ii) is model deterioration more related
to ET or to TWSA simulation?; (iii) does including a drought in the calibration period improve model transferability to severe
droughts?

For this purpose, we analysed the performance of the distributed hydrological model Continuum (Silvestro et al., 2013) over
the Po river basin in northern Italy and the flood- and drought-rich period September 2009 – August 2022. We calibrated the
model against Q data and we evaluated the modelling capabilities in reproducing the temporal and spatial variability of Q,





ET, and TWSA for the whole river basin and 38 sub-catchments during wet years and (moderate and severe) droughts, using
independent ground- and remote sensing-based datasets as benchmarks.

## 2    Data and methods

### 2.1    Study area

For this study, we selected the Po river basin, as a drought-prone area (Masante et al., 2017; Marchina et al., 2019; Toreti et al., 2022b, a), and major Italian catchment regarding drainage area (around 74000 km$^2$) and socio-economic relevance with the
27% of population, the 35% of agricultural production, and the 37% of industrial production of the whole country (Authority, 2006).

The Po river basin is located in the northern part of Italy and part of the Swiss Canton Ticino region (Figure 1). The basin is bounded by the Alps in the west and north, and the Apennines in the south, while the central part is characterized by the Po plain. Consequently, it shows a steep orographic gradient and elevations range from sea level to about 4800 m a.s.l. (Verdin,
2017) (Figure 1a).

The climate in the region shows a transition from alpine and cold, with a bimodal precipitation annual cycle and peaks in autumn and spring, to temperate with a dry season and most of the precipitation falling in winter (Beck et al., 2018; Crespi et al., 2018) (Figure 1b). Snow contribution to streamflow generation can be relevant, especially at high elevations in the northern and western part of the catchment, where the mean annual ratio between peak snow water equivalent and the annual cumulative Q
can exceed 60% (Avanzi et al., 2022). Subsequently, Q usually has two peaks, one in autumn caused by heavy rainfall events and one in spring caused by rainfall events and snowmelt, with a low-flow period during summer.

As a result of topographic and climatic characteristics, a variety of land cover types can be found in the basin (Figure 1c): transitions between bare soil, grassland, and forests following the elevational gradient in the mountainous parts, shrubland in the temperate and dry areas in the southwestern part, and cultivated and urban areas in the central lowlands (ESA, 2017). In
addition to three major lakes (Como, Maggiore, and Garda), the hydrographic system in the basin is influenced by around 180 multi-purpose reservoirs (Authority, 2006). Further, the hydrological cycle in the area is heavily affected by anthropogenic water withdrawals for irrigation, industrial, and drinking water uses. Irrigation accounts most among the water uses. Water withdrawals for irrigation are estimated at around 17*10$^9$ m$^3$/year (i.e., around the 5% of mean annual precipitation) and they further increase by up a 15% during droughts (Authority, 2006).

### 2.2    Hydrological modelling

The hydrological model Continuum (Silvestro et al., 2013) is a continuous grid-based hydrological model. It simulates the main hydrological processes in a process-oriented but parsimonious way, with only few calibration parameters, by solving both the mass and energy balances (Silvestro et al., 2013). The model includes optional modules to simulate lakes, dams, and hydraulic



infrastructures such as point water withdrawals and releases. Model code together with pre-processing tools is available here
https://github.com/c-hydro.

The model setup used here consisted of six modules (namely, the snow, vegetation, energy balance, soil, groundwater, and surface water modules) to simulate snow dynamics, vegetation interception, energy fluxes, evaporation from canopy layer, evapotranspiration, soil moisture and groundwater dynamics, and streamflow generation (Figure S1). Further, we simulated major lakes and dams in the region (Alfieri et al., 2022). We refer the reader to Silvestro et al. (2013) for a description of the
model, Silvestro et al. (2015) for specifics on the snow module, and Silvestro et al. (2021) for specifics on the surface flow routing scheme. In Figure S1 we provided a scheme of the model configuration, along with model output and states.

In this work, we run Continuum at a 0.009° (around 1 km) spatial resolution and 1 hour temporal resolution (Alfieri et al., 2022) over the hydrological years (h.y.) 2009 - 2022, with the first h.y. as warm-up period. Please note that throughout the manuscript we referred to hydrological year rather than calendar year and we considered it spanning from September to
August.

### 2.3   Data

#### 2.3.1   Model input data

In this work, we used the model setup and the required datasets, regarding topography, soil properties, land cover, dams, lakes, and glacier cover, presented in Alfieri et al. (2022). We summarized the input data in Table S1 and we refer the reader to Alfieri
et al. (2022) for a detailed description of them.

Further, dynamic maps of meteorological data - including precipitation (P), air temperature, relative humidity, wind speed, and shortwave solar radiation - are needed as model input data. For precipitation we used interpolated maps from ground- and radar-based P data from the Italian Civil Protection Department (in the following DPC), computed with the Modified Conditional Merging algorithm (Bruno et al., 2021). For other meteorological variables we used interpolated maps based on
ground-based data provided by DPC (see Alfieri et al. (2022) for details).

#### 2.3.2   Data for model calibration and evaluation

For model calibration and evaluation, we exploited a set of independent ground- and remote sensing-based datasets (Table S1). As Q dataset, we used quality-checked daily mean Q time series for 38 sub-catchments in the Po river basin (Figure 1) from DPC and Italian regional hydrometeorological offices. We selected the study sub-catchments according to Q data
availability (maximum 6 months of missing data). These sub-catchments reflect the variety of topographic, climatic, and land cover characteristics in the study area (Table S2).

For ET, we exploited the METv2 product by the Land Surface Analysis of the EUMETSAT Satellite Application Facility, in the following LSASAF product (Ghilain et al., 2011; EUM, 2016). The LSASAF product provides gridded ET estimates at a resolution of around 5 km over Europe and at a temporal resolution of 1 hour. The ET estimates are derived through a surface
energy model fed by remote-sensed data. This product showed reasonable agreement with alternative gridded ET products





and eddy-covariance data over Italy (Bruno et al., 2022). We used the LSASAF product as benchmark of simulated ET for both catchment-scale and spatial patterns analyses. Finally, we employed TWSA data from the Gravity Recovery And Climate Experiment (GRACE) and GRACE Follow-On (GRACE-FO) missions (henceforth GRACE data). GRACE was launched in April 2002 and dismissed in Jun 2017, whereas GRACE-FO is operational since May 2018. These satellite missions consist of

two twin satellites measuring variations in distance between them and, thus, in the Earth's gravity field. Consequently, GRACE data provide estimates of changes in mass over a certain area that can be mainly attributed to variations in Terrestrial Water Storage (TWS), i.e., in the groundwater, soil moisture, surface water bodies, snow, and ice storages. As GRACE data, we used the recently developed mass concentration (MASCON) solution, as it is particularly suited for hydrological applications compared to the traditional spherical harmonics solution (Scanlon et al., 2016). MASCON does not require any significant

postprocessing, while minimizing errors due to the leakage of the signal from land to oceans. We processed the latest products of GRACE MASCONS (release 06) provided by the Center for Space Research at the University of Texas (CSR) (Save et al., 2016; Save, 2020), the NASA Jet Propulsion Laboratory (JPL) (Watkins et al., 2015; Wiese et al., 2019), and the NASA Geodesy and Geophysics Research Laboratory (GSFC) (Loomis et al., 2019) at monthly temporal resolution, and spatial resolutions of $1°$ for CSR and GSFC products and $0.5°$ for the JPL product. We regridded the three products to a common

grid of $0.5°$ spatial resolution and then considered the mean among them to reduce the uncertainties associated with specific GRACE products (Scanlon et al., 2019). GRACE data are provided as anomalies regarding the period 2004 – 2009, therefore we converted them to anomalies about the study period by subtracting their long-term means (Scanlon et al., 2019). Due to the coarse spatial resolution of GRACE data and the relatively small drainage area for most of the study sub-catchments (Table S2), we used GRACE data only for catchment-scale analysis and at the outlet section of the basin (drainage area = 72545 $km^2$).

## 140 2.4 Analyses

### 2.4.1 Experimental design

We performed two calibration experiments differing in the calibration period; we then evaluated the model performances during two other intervals of the study period with contrasting climatic characteristics to evaluate the sensitivity of model performances to the climatic conditions either in the evaluation and calibration periods.

Specifically, we calibrated the model during "normal" years (2018 and 2019, mean annual P standardized anomaly according to Equation 2 for 2019 = 0.34±0.42 across the study sub-catchments, Figure 2) and during a moderate drought (2016 and 2017, mean annual P standardized anomaly = -0.85±0.61 for 2017, Figure 2). Then, we evaluated model performances in independent wet years (2014 and 2020, with mean annual P standardized anomaly across the study sub-catchments = 1.14±0.65 in 2014 and equal to 1.48±0.34 in 2020, Figure 2), moderate droughts (2012 and 2017, with mean annual P standardized anomaly

across the study sub- catchments = -0.8±0.39 in 2012 and equal to -0.85±0.61 in 2017, Figure 2), and a severe drought (2022, with mean annual P standardized anomaly across the study sub-catchments = -1.68±0.43, Figure 2).





### 2.4.2 Model calibration

We used a multi-site calibration procedure to calibrate the model against Q data at 18 sub-catchments over a 2-year period (first six months as model warm-up) following the approach of Alfieri et al. (2022). We calibrated four model parameters (Figure

S1): the Curve Number (CN), the field capacity ($c_t$), the infiltration velocity at saturation ($c_f$), and a parameter regulating the baseflow from the groundwater storage ($w_s$). CN, $c_t$, and $c_f$ are spatially distributed parameters obtained by rescaling global maps of soil characteristics (Poggio et al., 2021) and land cover (ESA, 2017), while $w_s$ is lumped for the whole model domain. As calibration procedure, we used a parallel search algorithm that iteratively explores the 4D parameter space until convergence (improvement in the cost function < 1% compared to previous iteration). In the cost function we used the Kling-

Gupta Efficiency (KGE, Kling et al. (2012)) on daily Q. The KGE is an aggregated measure of agreement in the timing, magnitude, and variability between simulations and observations, according to Equation 1:

$$KGE = 1 - \sqrt{(r-1)^2 + (\beta-1)^2 + (\gamma-1)^2} \tag{1}$$

where r is the Pearson's correlation coefficient, $\beta$ is the ratio between simulated and observed mean, and $\gamma$ is the ratio between the simulated and observed coefficient of variation (KGE $\in$ (-∞, 1], optimal value = 1, no-skill threshold over mean

flow as predictor = -0.41 as per Knoben et al. (2019)). Further, we weighted the KGE with the logarithm of the drainage area to give more emphasis to downstream sub-catchments Alfieri et al. (2022). We reported the KGE from the two calibration experiments in Table S2.

### 2.4.3 Model evaluation

We evaluated model performances in reproducing Q, ET, and TWSA temporal (and spatial) variability at monthly time scale, as

this is the temporal resolution of GRACE data. To evaluate model skills for TWSA, we reconstructed the simulated TWS from model storages states, i.e., from the water volumes in the snow ($S_{snow}$), vegetation ($S_{veg}$), surface water ($S_{sw}$), soil ($S_{soil}$), and groundwater ($S_{gw}$) storages (Figure S1). We then converted it to TWSA, i.e., deviations from the long-term TWS mean for the simulation period. For the catchment-scale analysis of both ET and TWSA we used catchment-average time series. Further, because of high seasonality in hydrological processes in the region, we evaluated model capabilities in simulating seasonality

(i.e., monthly mean values), deviations from it (i.e., monthly standardized anomalies), and long-term changes. We evaluated the model capabilities in simulating long-term changes only qualitatively, as we considered the study period too short for trend detection.

We computed the monthly standardized anomalies, $z_{anom}$ (hereafter anomalies for brevity) according to Equation 2:

$$z_{anom}(t_i) = \frac{z(t_i) - \overline{z_i}}{\sigma_{z_i}} \tag{2}$$

where z is the value at each time step, and $\overline{z_i}$ and $\sigma_{z_i}$ are the long-term mean and standard deviation for month i.





As performance metrics for model evaluation, we used the KGE (Section 2.4.2), the Pearson's correlation coefficient (r, with r ∈ [-1, 1] and 1 as optimal value), and the normalized Root Mean Square Error (nRMSE, with nRMSE ∈ [0, +∞) and 0 as optimal value Moriasi et al. (2007)), according to Equation 3:

$$nRMSE = \frac{\sqrt{\frac{1}{N}\sum_{i=1}^{N}(X_{sim,i} - X_{obs,i})^2}}{\sigma_{X_{obs}}} \tag{3}$$

where $X_{sim,i}$ is the simulated variable at time step i, $X_{obs,i}$ the observed, $\sigma_{X_{obs}}$ the observed standard deviation, and N the number of time steps. r is a measure of the agreement in timing, while nRMSE measures the general agreement between simulations and benchmark. We normalized the RMSE to allow comparison among sub-catchments/grid cells. For all the normalizations we used the standard deviation rather than the widely used mean to avoid numerical issues when the mean is close to zero as in the case of TWSA.

## 3 Results

### 3.1 Hydroclimatological conditions during droughts

Three droughts occurred in the region during the study period, namely in 2012, 2017, and 2022 - ongoing - as reported by Masante et al. (2017); Marchina et al. (2019); Toreti et al. (2022b, a). These three events were all characterized by a winter P deficit (Figure 3a). However, duration and severity of P deficits differed among the events. P deficits were moderate in 2012 and 2017, and severe in 2022 (Section 2.4.1, Figure 2). Furthermore, during the three events P deficits propagated rather differently through the hydrological cycle as revealed by Q, ET, and TWSA data for the basin outlet (Figure 3). For 2012 and 2017, the LSASAF product showed positive ET anomalies during spring (Figure 3b), but negative anomalies during summer (August ET = 52 mm month$^{-1}$ in 2012 and 46 mm month$^{-1}$ in 2017 compared to a climatology of 71±15 mm month$^{-1}$ over the study period 2010-2022). On the contrary, the ET product showed positive ET anomalies during summer for the 2022 event (Figure 3b, July ET = 124 mm month$^{-1}$ compared to a climatology of 87±18 mm month$^{-1}$). The 2022 drought was indeed associated with a summer heatwave (Toreti et al., 2022a) that may have contributed to positive ET anomalies over the study region. Further, in 2012 and 2017 TWSA was within the climatology for the whole hydrological year, whereas it was already low at the beginning of the 2022 h.y. (Figure 3c, September TWSA = -92 mm compared to a climatology of -58±37 mm) and it reached an historical minimum value during summer (August TWSA = -158 mm compared to a climatology of-54±56 mm, Figure 3c). As a results, Q showed mild negative anomalies throughout the hydrological years 2012 and 2017 (Figure 3d, July Q = 18 mm in 2012 and 25 mm in 2017, compared to a climatology of 30±13 mm), while it experienced strong negative anomalies during most of 2022 (Figure 3d, July Q = 9 mm in 2022).





### 3.2 Model evaluation for streamflow during droughts of different severity

The model simulated Q comparably well during wet years and moderate droughts, with mean KGE values for monthly Q

across the study sub-catchments equal to 0.59 during 2014 and 2020, and equal to 0.55 during 2012 and 2017 (non-significant difference according to a t-test for the mean, pvalue > 0.01, Figure 4a, b, and d). Q simulation was skilful even during the severe 2022 drought when considering the basin outlet, which was assigned the highest weight in calibration (KGE = 0.82). At the basin outlet, the model properly represented the slight decline in Q since autumn 2019 (24-month running means in Figure 5a) and Q seasonality, with r = 0.91 and nRMSE = 0.48 for monthly mean Q (Figure 5b). Nonetheless, for 2022 the model had a

drop in performances across all other sub-catchments (mean KGE = 0.18±0.69, statistically different mean KGE compared to those for wet years and moderate droughts according to a two-sample t-test, pvalue < 0.01), even though the model preserved some skills over a climatological mean (Knoben et al., 2019). Modelling skills were low especially in western catchments, which experienced negative P anomalies already in 2021 and the strongest negative P anomalies across the basin in 2022 (Figure 1). In other words, the drop during the severe 2022 drought was larger in those sub-catchments where the 2022 drought

was especially severe and prolonged. In the following we investigated and discussed possible causes for the deterioration of Q simulation in those catchments during 2022, i.e., ET and TWSA simulation (Section 3.3) and the human disturbance on Q during severe droughts (Section 4.2).

### 3.3 Model evaluation for evapotranspiration and terrestrial water storage anomalies

To identify possible causes for the drop in Q modelling skills during the severe 2022 drought across most of the sub-catchments

(Section 3.2), we evaluated model skills for ET and TWSA.

Integrated over the entire basin, the model properly simulated both ET monthly values (Figure 5d) with r = 0.94 and nRMSE = 0.36, and ET seasonality (Figure 5e), with r = 0.99 and nRMSE = 0.18 for monthly mean ET. However, the model slightly overestimated ET during winter and spring, and it simulated an earlier ET peak in summer (Figure 5e). Moreover, the model had rather low performances in simulating ET deviations from seasonality, with r = 0.52 and nRMSE = 0.98 for monthly ET

anomalies (Figure 5f), which further deteriorated during the severe drought (r = 0.07 and nRMSE = 1.69 for 2022) compared to moderate droughts (r = 0.89 and nRMSE = 0.5 for 2012 and 2017).

Regarding TWSA over the entire basin, the model properly simulated the declining trend since about autumn 2019 (visualized by the 24-month running means in Figure 5g) and TWSA seasonality (r = 0.91 and nRMSE = 0.41, Figure 5h). Model skills in reproducing TWSA monthly values are comparatively lower than those in simulating Q and ET, with r = 0.76 and

nRMSE = 0.68 (Figure 5g). The model had comparatively poor performances in simulating TWSA deviations from seasonality, with r = 0.66 and nRMSE = 0.81 for monthly TWSA anomalies (Figure 5i). Yet, model performances in simulating TWSA anomalies were comparable during moderate and severe droughts, with r = 0.41 and nRMSE = 1.41 during 2012 and 2017, and r = 0.67 and nRMSE = 1.73 during 2022 (Figure 5i). Therefore, we argue that the drop in Q modelling skills for most sub-catchments in 2022 was related to the representation of ET – and to ET anomalies in particular – rather than TWSA.





Indeed, the simulation of ET anomalies across the study sub-catchments was skillful during moderate droughts (mean r = 0.81 and mean nRMSE = 0.68, Figure 6a and d), but for most of the sub-catchments it deteriorated significantly during the severe drought (mean r = 0.05 and mean nRMSE = 1.61, Figure 6b and e), with statistically different mean compared to those during moderate droughts according to a two-sample t-test (pvalue < 0.01, Figure 6c and f).

Performance drops for ET anomalies during the severe drought were not uniform throughout the model domain. They were

particularly pronounced in the central part of the model domain (Figure 7a, b, d, and e) and showed a clear pattern with land cover. Model deterioration was particularly strong for croplands, which are mostly situated in the central part of the model domain (Figure 1c), with mean r = 0.58 and mean nRMSE = 0.97 across the model cells classified as crop during moderate droughts, compared to r = -0.03 and mean nRMSE = 1.8 during the severe drought (Figure 7c and f).

### 3.4    Impact of calibration period on model transferability

Including a moderate drought (the 2017 event) in the calibration period did not lead to an improvement in modelling skills during the severe drought (2022). Model performance during calibration was similar during both calibration experiments – with a mean KGE across the calibrated sub-catchments = 0.58 for the "normal" calibration period and a mean KGE = 0.44 for the calibration period including a moderate drought (not statistically different mean according to a two-sample t-test, pvalue > 0.01, Table S2). Also for the model calibrated during a drought model performances in simulating monthly Q across the study

sub-catchments deteriorated for the severe 2022 drought compared to model skills during moderate droughts (mean KGE = 0.5 during moderate droughts and mean KGE = 0.18 during the severe drought, statistically different according to two-sample t-test with pvalue < 0.01, Figure 8c).

The model calibrated during a moderate drought showed similar issues in simulating ET anomalies in the croplands during a severe drought as the model calibrated during "normal" years (for croplands mean r = 0.59 during moderate droughts and

mean r = -0.11 during the severe drought, Figure 8f, while mean nRMSE = 0.97 during moderate droughts and mean nRMSE = 1.85 during the severe drought, Figure 8i).

## 4    Discussion

### 4.1    Main findings and implications

We investigated Q modelling skills during moderate and severe droughts for a distributed process-based hydrological model,

we explored ET and TWSA simulation as a possible cause for the expected drop in model performances, and we evaluated the benefit of including a moderate drought in the calibration. Our findings in this regard were three.

First, Continuum represented reasonably well Q also during moderate droughts such as the 2012 and 2017 events over the Po River basin in Italy (KGE = 0.55±0.25 across the 38 study sub-catchments in the basin and KGE = 0.77 integrated over the entire basin). However, we also found that a severe drought like the 2022 event could challenge Q simulation, with a decrease

in model performances across the study sub-catchments (KGE = 0.18±0.69), even though the model reliably simulated Q



integrated over the entire basin also in 2022 (KGE = 0.82). Deb and Kiem (2020) tested the ability of three hydrological models (lumped, semi-distributed, and distributed) to simulate Q outside the climatic conditions of the calibration period for two catchments in South-eastern Australia and they found better performances for most of the hydro-climatic conditions from the distributed model. However, a wide number of studies reported drops in Q modelling skills when simulating prolonged dry
periods and particularly severe Q deficits, such as the Millennium Drought in Australia (Saft et al., 2016) and the Californian multi-year drought between 2012 and 2016 (Avanzi et al., 2020). Therefore, our results showed, on one hand, the ability of the Continuum hydrological model in simulating Q even during moderate droughts and, on the other hand, the need for research investigating the causes for drops in Q modelling capabilities during severe droughts.

Second, we argued that the drop in Q modelling performances during the severe 2022 drought event can be related to the mis-
representation of ET anomalies, among other factors (discussed in Section 4.2). Here, the model properly represented ET and TWSA seasonality and long-term variability, while model performances slightly decreased for their deviations from seasonality, coherently with previous literature. Bolaños Chavarría et al. (2022) for instance showed that a set of global hydrological and land surface models well represented TWSA seasonality and long-term variability in a tropical river basin in Colombia, but not the TWSA monthly time series that account for the deviations from seasonality. However, model capabilities in sim-
ulating TWSA anomalies were comparable during moderate droughts and a severe drought. On the contrary, we showed here that model capabilities in simulating spatial and temporal variability of ET anomalies decreased significantly during a severe drought, especially in the croplands, compared to moderate droughts. Previous studies have shown that Q simulation during droughts can be hampered by a poor simulation of ET (Avanzi et al., 2020) that can have a prominent role during severe and prolonged drought events (Brunner et al., 2022; Massari et al., 2022).

Third, including a moderate drought (the 2017 event) in the calibration did not lead to an improvement in Q and ET simulation during a severe drought (the 2022 event), with mean KGE = 0.18 for Q across the study sub-catchments, and mean r = -0.11 and nRMSE = 1.85 for ET across the croplands during 2022. Yang et al. (2021) reported that an ecohydrological model better simulated Q in an experimental German catchment during the 2018-2019 drought when including it in the calibration period. However, here we proved that calibrating during a moderate drought was not sufficient to improve model transferability
to a different and more severe drought.

While we demonstrated the ability of a distributed and process-oriented hydrological model in simulating Q, ET, and TWSA during moderate droughts, our results pointed to a broader need for better representing drought and human processes in hydrological models to achieve robust simulation also during severe droughts. Recent literature revealed that a changing climate may exacerbate the occurrence of severe and prolonged drought events (Rakovec et al., 2022). Thus, comprehensive evaluations
of simulated hydrological fluxes and states, and testing alternative strategies to enhance the simulation of the hydrological cycle during severe droughts are warranted whenever a hydrological model is used with specific focus on droughts, such as in drought studies, drought monitoring and forecasting systems, and impact assessments of a drying climate.



## 4.2 Future work

Our study area encompassed a variety of climates and land cover types (Figure 1) and our study period included moderate and
severe droughts (Figure 3). However, our results referred to a particular region and specific drought events. Hence, studies over
different areas and droughts would be helpful to generalize our conclusions.

In this work, we showed the value of remote sensing-based products to benchmark simulated ET and TWSA, especially for
spatial patterns analyses. However, ET and TWSA retrieval through remote sensing still presents challenges, as ET cannot be
directly measured and TWSA can be derived only at large scales. For TWSA, we based on the latest GRACE products (Section
2.3.2) and we used the mean from three products to take into account their uncertainty (Scanlon et al., 2019). As ET dataset,
we exploited the LSASAF product which showed skilful performances over the study region, even during droughts (Bruno
et al., 2022). Benchmarking the model against additional remote sensing-based datasets, for ET or additional variables such as
soil moisture, would be beneficial to further verify model internal consistency during droughts.

Multivariable calibration may be helpful to improve model internal consistency (Dembélé et al., 2020; Duethmann et al.,
2022), also during low-flow periods (Rakovec et al., 2016a) and droughts (Yang et al., 2021). Yang et al. (2021) for example
showed that including tracer data in the calibration of an ecohydrological model increased model internal consistency dur-
ing the 2018-2019 drought in Central Europe. Here we calibrated the model against Q data only (Section 2.4.2). Given the
satisfactory performances we achieved for ET during moderate droughts, we argue that a multi-variable calibration approach
would not significantly enhance model transferability to a severe drought. Further, Hartmann and Bárdossy (2005) showed that
a multi-objective calibration with Q data aggregated at different time scales improved Q transferability outside the calibration
conditions for a distributed model in a German medium-sized basin. A similar approach could be tested in future work.

The hydrological cycle in our study area is heavily affected by human interference, both in terms of water withdrawals and
irrigation (Section 2.1). Here, the Continuum hydrological model included reservoirs - although their management was not
known -, but not irrigation and water withdrawals that can be especially relevant during droughts rather than during wet pe-
riods. By calibrating the model against observed Q data, we partly considered human interference in model parameterization.
However, irrigation – which can strongly increase water availability for ET - might be one of the reasons for the mismatch
between simulated and remote sensing-based ET spatial variability we detected during the severe 2022 drought. Therefore, an
enhanced representation of human interference, in terms of either data assimilation or model structure, could improve hydro-
logical modelling during severe droughts. For instance, Rameshwaran et al. (2022) achieved a median 10.6% improvement in
low-flows simulation by including monthly actual abstraction and discharge data in a distributed hydrological model for 605
English catchments. Further, Mocko et al. (2021) showed that assimilating vegetation variables into a land surface model leads
to an improved simulation of agricultural droughts and Dari et al. (2022) proposed effective techniques for estimating irriga-
tion over large areas through satellite data that can be incorporated into distributed hydrological modelling. Further research is
needed to investigate the benefits of data assimilation in the general representation of the hydrological cycle including human
interference during severe droughts.





## 5   Conclusions

In this work, we comprehensively evaluated model capabilities in reproducing Q, ET, and TWSA during droughts of different severity for the distributed hydrological model Continuum over 38 sub-catchments of the Po River basin in northern Italy, basing on ground- and remote sensing-based datasets as independent benchmarks. Further, we tested the value of calibrating
during a moderate drought as possible strategy to improve model performances during a severe drought. We found that the distributed hydrological model represented well Q during moderate droughts (i.e., the 2012 and 2017 events) even in a highly human-affected area, but not during a severe drought like the 2022 event for many of the study sub-catchments (Figure 4). We linked this drop in modelling performances to a misrepresentation of ET anomalies in the irrigated croplands during such period (Figure 7). Further, we demonstrated that issues in properly representing Q and ET during a severe drought were not
sensitive to the climatic conditions in the period used for calibration (Figure 8). Thus, holistic model evaluations for the different components of the hydrological cycle and possibly model developments to enhance the representation of human interference, also through the inclusion of new data, are needed to increase model robustness during severe droughts. This is highly relevant in a changing climate and the anthropogenic era to properly predict water availability throughout the hydrological cycle, and inform water management, disaster risk reduction, and climate change adaptation measures.

*Code and data availability.*   Sources of data used in this work were reported in Table S1. Model code and preprocessing tools are available at https://github.com/c-hydro.

*Author contributions.*   GB, DD, FA, and SG conceptualized the study. GB performed the analyses and developed the methodology and the software with help from LA and AL. All authors discussed and contributed to the interpretation of results. GB prepared the data visualizations and the manuscript with inputs from all authors.

*Competing interests.*   The authors declare that they have no conflict of interest.

*Acknowledgements.*   This paper has been supported by the Italian Department of Civil Protection, Presidency of the Council of Ministers, through the convention between Department of Civil Protection and CIMA Research Foundation, for the development of knowledge, methodologies, technologies and training useful for the implementation of national systems of monitoring, prevention and surveillance.





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

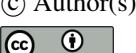



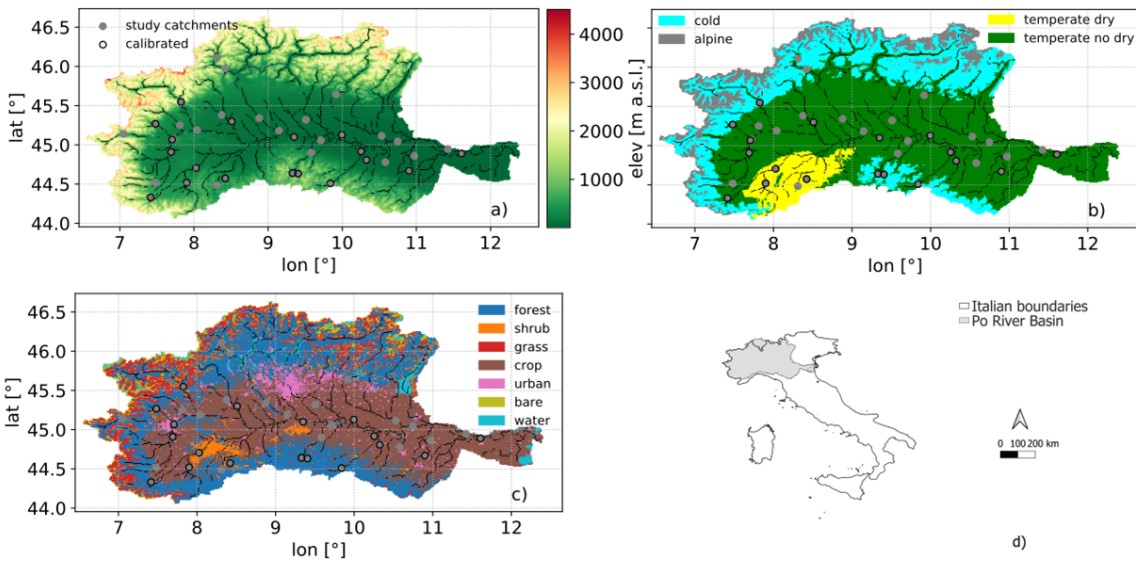

**Figure 1.** Maps with (a) elevation, (b) climates, (c) land cover types, and (d) location of the model domain, modelled river network (black line), and study catchments (grey dots and black edge if calibrated). For data sources please refer to Table S1.

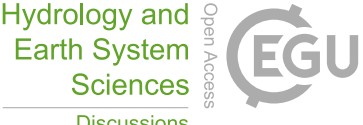

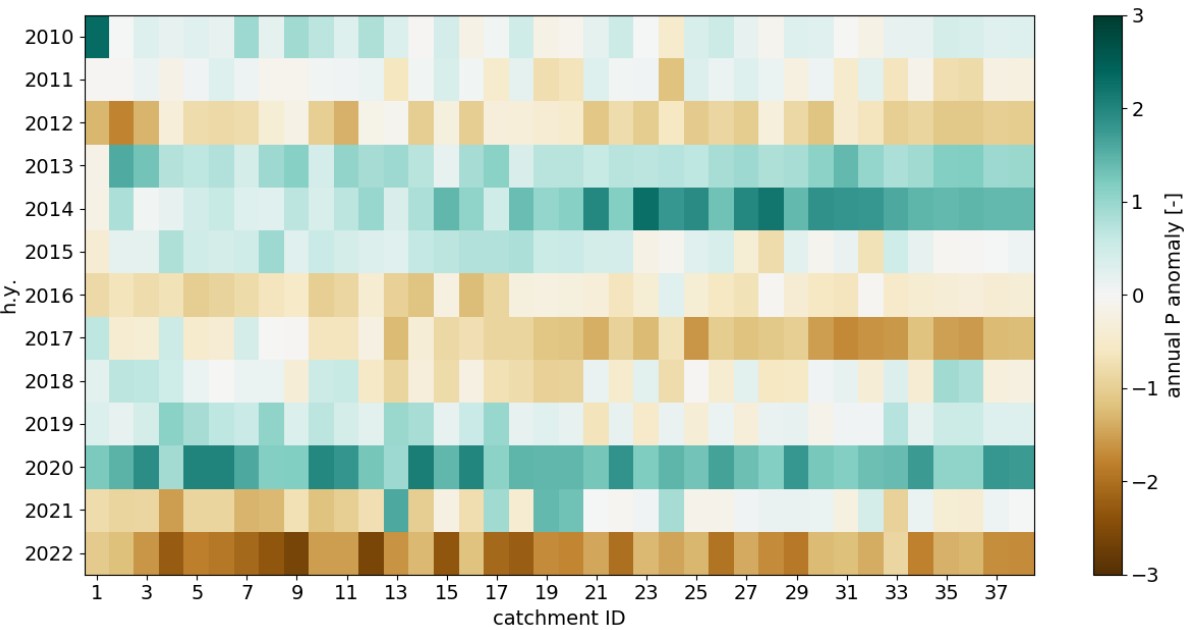

**Figure 2.** Annual P standardized anomalies for each study sub-catchment (west to east ordered, from the left to the right end side) over the study period. We computed standardized anomalies according to Equation 2.

**Figure 3.** P (a), ET (b), TWSA (c), and Q (d) observed monthly climatology (mean and standard deviations over 2010-2022) and monthly values during drought years, for the basin outlet.





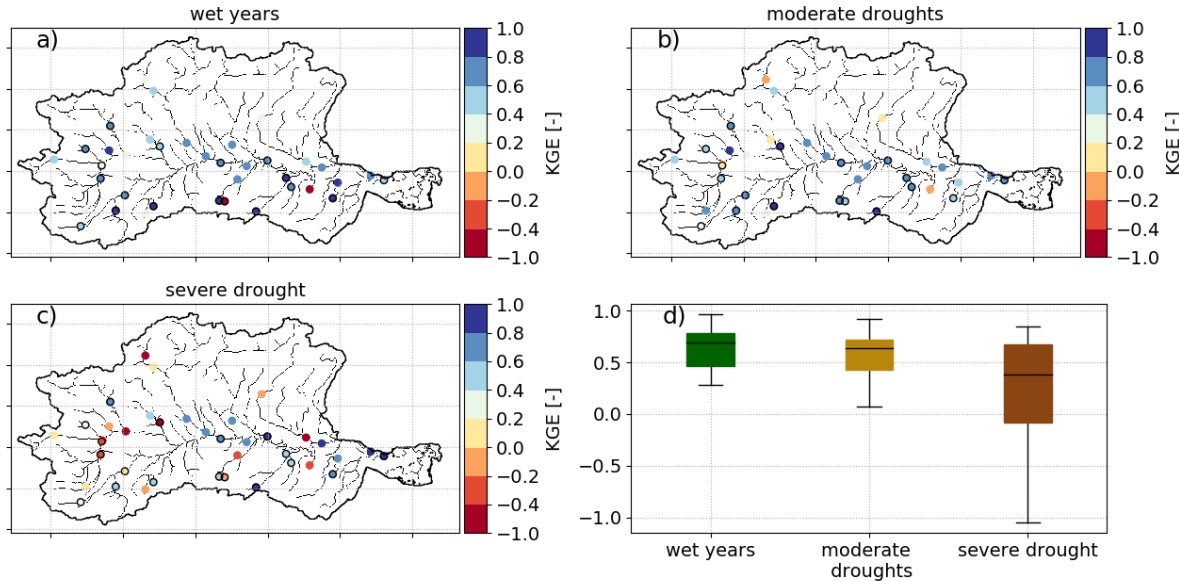

**Figure 4.** KGE on monthly Q during wet years (a), moderate droughts (b), and the severe drought (c) for each study sub-catchment, and KGE distributions as boxplots (d).



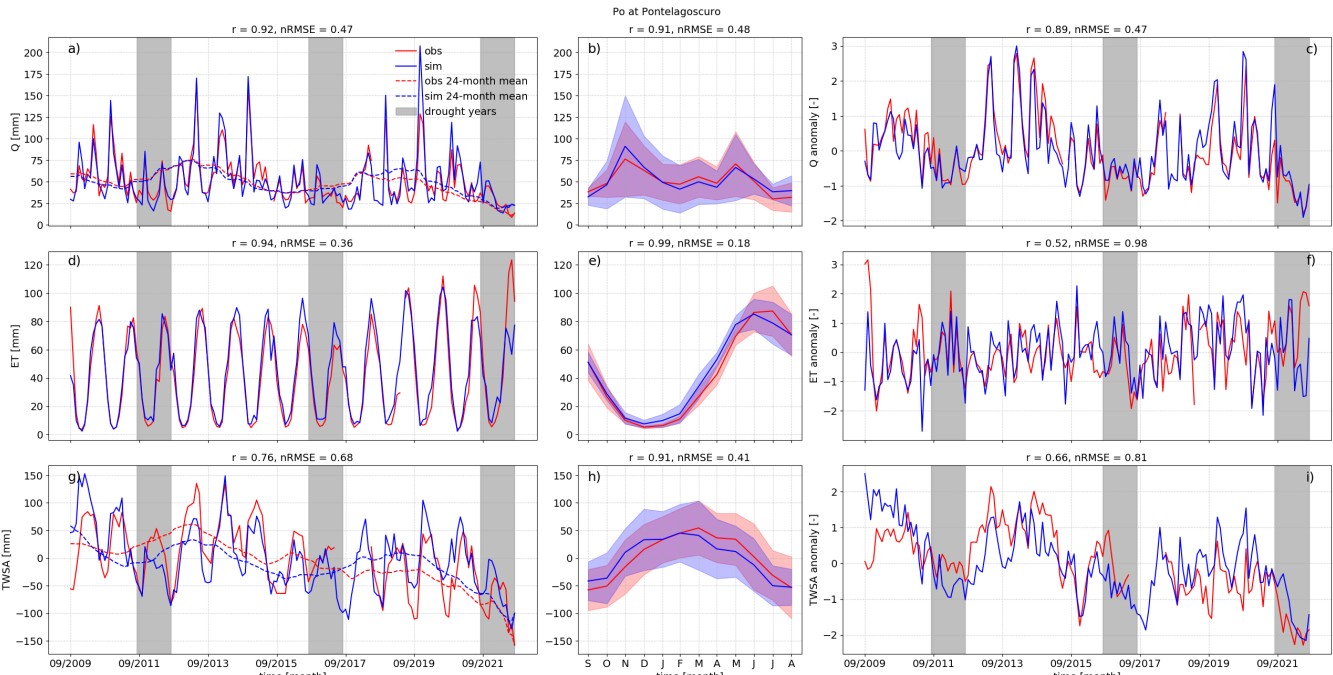

**Figure 5.** Time series of observed (red) and simulated (blue) Q (first row), ET (second row), and TWSA (third row) monthly values with 24-month rolling means (first column), monthly means (second column), and monthly standardized anomalies (third column) for the basin outlet. Shading in panels b, e, and h corresponds to ± 1 standard deviation.





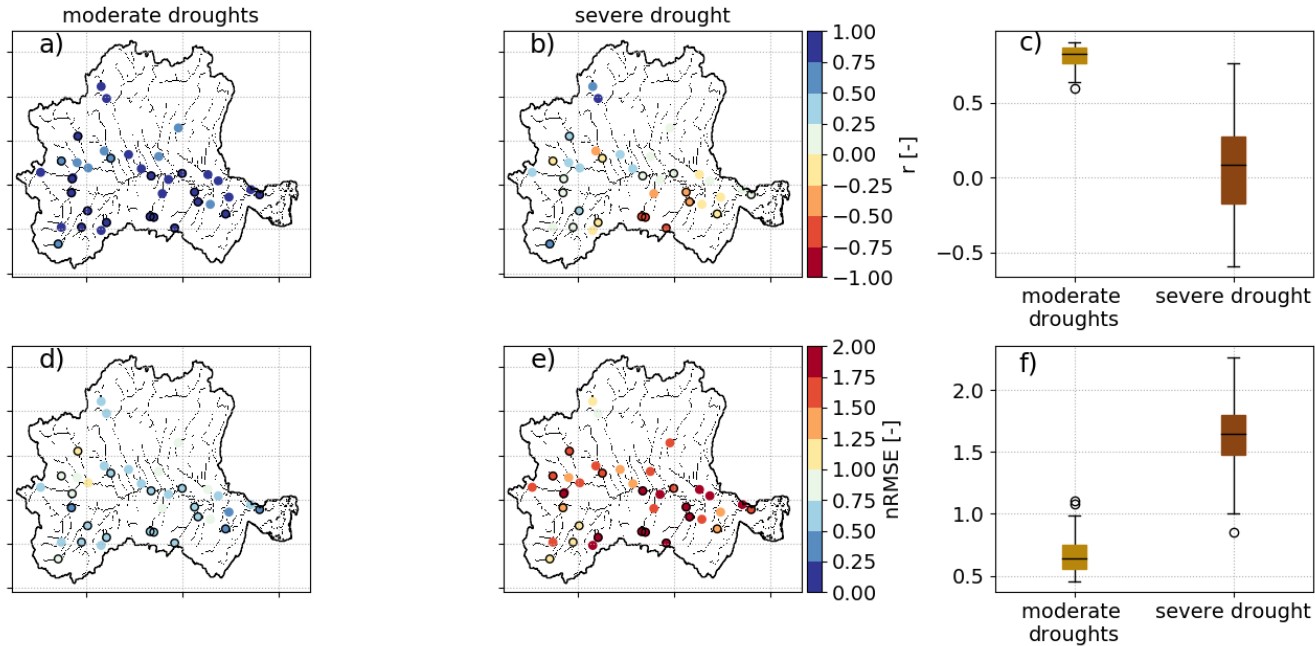

**Figure 6.** r and nRMSE on monthly ET anomalies over moderate droughts (a and d) and a severe drought (b and e) for each study sub-catchment, and errors distributions as boxplots (c and f).



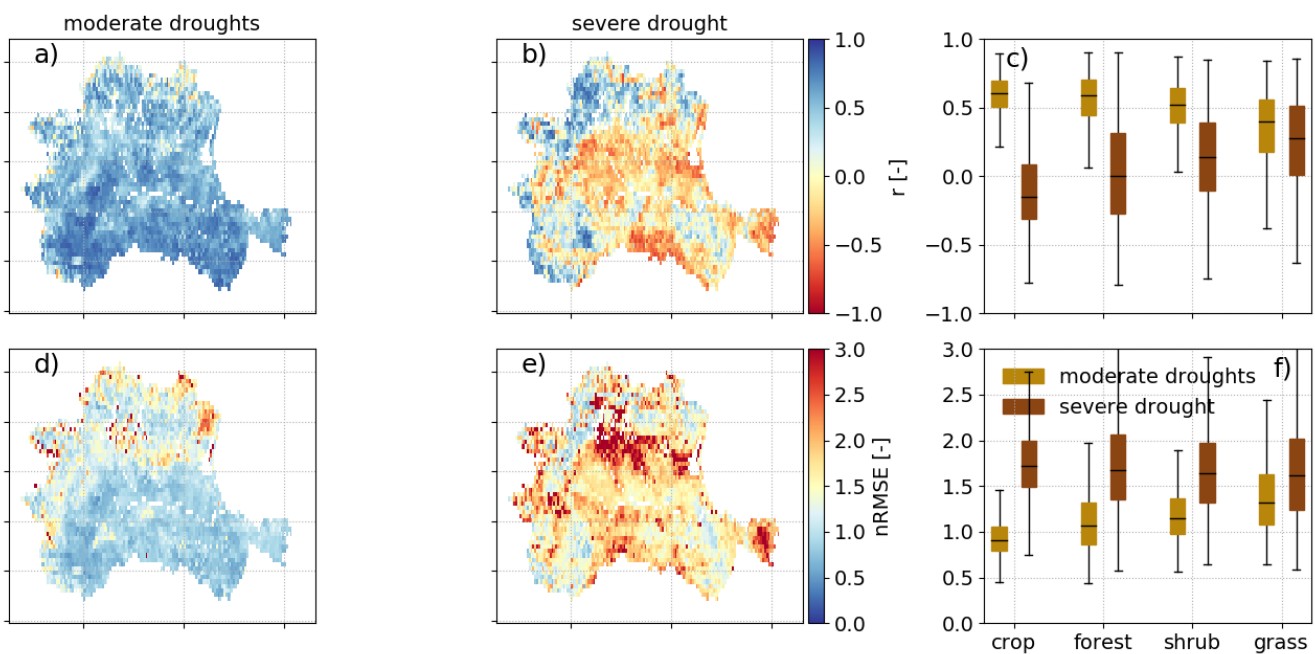

**Figure 7.** Maps of pixel-wise r and nRMSE on monthly ET anomalies over moderate droughts (a and d) and the severe drought (b and e), and errors distributions as boxplots per each land cover type in the model domain (c and f). Water bodies were excluded from the comparison. Model outputs were rescaled by bilinear interpolation to the resolution of the LSASAF product for comparison.





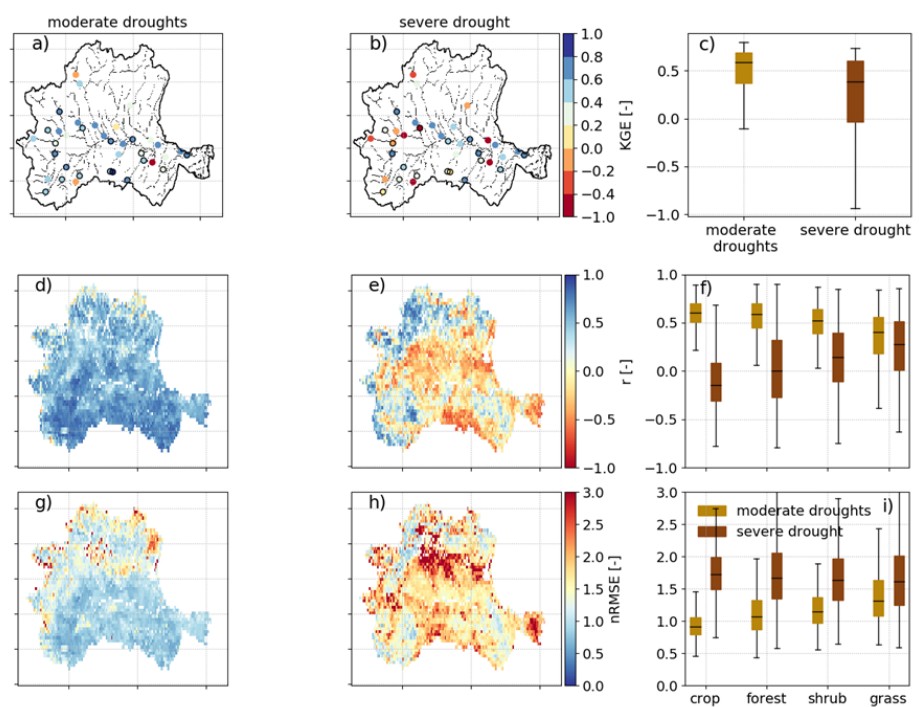

**Figure 8.** KGE on monthly Q over moderate droughts (a) and the severe drought (b) for each study sub-catchment, and KGE distributions as boxplots (c) from the model calibrated during a drought. Maps of r and nRMSE on monthly ET anomalies over moderate droughts (d and g) and the severe drought (e and h), and errors distributions as boxplots per each land cover types in the model domain (f and i) from the model calibrated during a drought.