# Peer review of "Parameter transferability of a distributed hydrological model to droughts"

_Hydrology and Earth System Sciences, 2022_

## Referee Comment (RC1)

This manuscript presents different calibrations and evaluations based on different climatic conditions spatially and temporally intended to explore how the model performance are sensitive to drought condition and the real potential causes data, and test that a drought included in calibration period would improve model transferability or not. The authors designed two calibration experiments and three evaluations under three different wetness conditions. They mainly found that a drop in performance of Q indeed happened in their study area based on Continuum model, and was related to the representation of ET anomalies rather than TWSA, and including a moderate drought in the calibration did not lead to an improvement in Q and ET simulation during a severe drought. The research makes a contribution to understanding the application of LSASAF product, and model performance under different climatic conditions.

However, I have some major concerns that need to be addressed prior to reconsideration:

1) Based on the results, I don't think authors can say that "the drop in Q modelling performances during the severe 2022 drought event can be related to the mis-representation of ET anomalies, among other factors". I just observed that a drop happened in ET performance from moderate droughts to severe droughts which was similar with that for Q performance. I didn't see any other evidence to prove that drop in Q performance can be related to ET simulation. Please design more experiments to give audience more evidence.

2) I don't think the authors really solved the research question: would a drought in calibration period improve model transferability or not? Authors just have two calibration experiments( one with normal period and another one with moderate drought) and then compared the evaluated results in severe drought. The results in this study were very different from that in Yang et al. (2021). But this may result from the different model which Yang used and this reference can not prove your result is correct. I suggested that author can design more experiments including different type of droughts based on different models and compare their results so that make your results more reliable.

3) how do you define wet, normal, moderate drought, and severe drought? I didn't see detailed clarification or an indicator in this manuscript.

4) could you please add the evaluation performance results in supplementary?

5) please make your paragraph format consistency.

6) what does the grey shade represent in Figure 5? Please add that in the text blow the figure.

7) what does the river basin really look like? When I see the river basin in figure1 and 4, the river basin looks well, however, the river basin in figure6-8 looks like that it was stretched vertically. Please make the river basin consistency in your figures. And please organize your figure6-8 better.

8)Line 203, what is "a climatology"? please clarify it in details.

---

## Author Comment (AC1)

Dear Referee #1,

We would like to thank you for your review of our paper and your fruitful comments on it. Please find below our replies (plain text) to your comments (in italic) with intended changes to the manuscript.

*This manuscript presents different calibrations and evaluations based on different climatic conditions spatially and temporally intended to explore how the model performance are sensitive to drought condition and the real potential causes data, and test that a drought included in calibration period would improve model transferability or not. The authors designed two calibration experiments and three evaluations under three different wetness conditions. They mainly found that a drop in performance of Q indeed happened in their study area based on Continuum model, and was related to the representation of ET anomalies rather than TWSA, and including a moderate drought in the calibration did not lead to an improvement in Q and ET simulation during a severe drought. The research makes a contribution to understanding the application of LSASAF product, and model performance under different climatic conditions.*

*However, I have some major concerns that need to be addressed prior to reconsideration:*

*1) Based on the results, I don't think authors can say that "the drop in Q modelling performances during the severe 2022 drought event can be related to the mis-representation of ET anomalies, among other factors". I just observed that a drop happened in ET performance from moderate droughts to severe droughts which was similar with that for Q performance. I didn't see any other evidence to prove that drop in Q performance can be related to ET simulation. Please design more experiments to give audience more evidence.*

The drop in the simulation of streamflow (Q) during the severe drought was mainly driven by an overestimation of Q in evaluation sub-catchments. To clarify this point, we will adjust Figure 4 by grouping the boxplots in panel d into calibration and evaluation sub-catchments, and we will add in the supplement similar figures for the components of the Kling Gupta Efficiency (correlation, bias, and variability). We will add further analyses to explore the potential causes for this. Specifically, we will investigate (i) systematic biases in observed data that may have occurred during the severe drought, because of increased uncertainty in P data or enhanced human disturbance on Q data for instance, (ii) the overestimation of the simulated contribution of Terrestrial Water Storage (TWS) to Q generation, and (iii) the underestimation of simulated evapotranspiration (ET). The uncertainty in observed data we used to force and evaluate the model has not increased systematically during the severe drought (69±234 mm in 2012 and 51±202 mm in 2022 as mean ± 1 standard deviation across the study sub-catchments). Furthermore, at the outlet section the model slightly overestimated TWS during the severe event (simulated TWS = -65 mm vs observed TWS = -92 mm in September 2021, and simulated TWS = -100 mm vs observed TWS = -158 mm in August 2022, Figure 5), and thus it did not overestimate its contribution to Q. Therefore, we conclude that the underestimation of ET was the main cause for the drop in Q performances, especially for human disturbed areas, as emerged also from the analysis on spatial patterns. Previous literature on the topic supports our finding. Avanzi et al. (2020) for instance identified the misrepresentation of ET elasticity to climate variability as the culprit for the drop in Q simulation during the 2012-2016 Californian drought for a semi-distributed hydrological model. We will revise Sections 3.2, 3.3, and 4.1 extensively to show and discuss these points.

*2) I don't think the authors really solved the research question: would a drought in calibration period improve model transferability or not? Authors just have two calibration experiments( one with normal period and another one with moderate drought) and then compared the evaluated results in severe drought. The results in this study were very different from that in Yang et al. (2021). But this may result from the different model which Yang used and this reference can not prove your result is correct. I suggested that author can design more experiments including different type of droughts based on different models and compare their results so that make your results more reliable.*

Yang et al. (2021) tested different calibration strategies for the simulation of the 2018-2019 German drought with an ecohydrological model in an experimental catchment and they reported an improvement in model performances by including the drought in the calibration period, compared to those from a wet calibration period. However, Avanzi et al. (2020) revealed that a semi-distributed hydrological model calibrated also during a drought had a drop in model performance when evaluating it during the 2012-2016 Californian drought. Here we calibrated a distributed hydrological model during a moderate drought (the 2017 event over the Po river basin), and then evaluated it during an independent and more severe drought (the 2022 event), without substantial improvements in model performances compared to those from an alternative calibration period. We agree that our conclusions differ from those in Yang et al. (2021) and, in our discussion, we indeed intended to refer to Yang et al. (2021) as a study contrasting our conclusions, rather than supporting them. This may be due to a number of differences between the two studies: first of all the experimental design (use of an independent drought as period for model evaluation), as well as differences in models, study areas, and calibration procedures used. We will clarify this by expanding lines 292-295 in Section 4.1. We agree that comparing different models in their transferability to severe droughts when calibrated during moderate droughts could provide interesting insights on the topic. While this is beyond the scope of the current paper, we will add it as a further possible way forward for future research in Section 4.2.

*3) how do you define wet, normal, moderate drought, and severe drought? I didn't see detailed clarification or an indicator in this manuscript.*

We characterized the different wetness conditions over the study period in terms of annual P standardized anomalies, as reported in Section 2.4.1 and Figure 2. We identified the wet/dry periods as periods with positive/negative anomalies for most of the study sub-catchments. Further, we referred to dry years as droughts, and we defined them moderate and severe in terms of decreasing annual P standardized anomalies. We did not set any specific threshold on annual P standardized anomalies to define drought years and characterize their severity; however, our drought characterization agrees with previous literature and drought reports (Masante et al., 2017; Marchina et al., 2019; Toreti et al., 2022a, b). We will clarify this in Section 2.4.1.

*4) could you please add the evaluation performance results in supplementary?*

Yes, we will add the evaluation scores for Q over all the evaluation periods for each calibration experiment and sub-catchment in a table in the supplement material.

*5) please make your paragraph format consistency.*

We will make our paragraph format consistent throughout the manuscript.

*6) what does the grey shade represent in Figure 5? Please add that in the text blow the figure.*

The grey shade in Figure 5 represents the analyzed drought years and we will specify it in the caption.

*7) what does the river basin really look like? When I see the river basin in figure1 and 4, the river basin looks well, however, the river basin in figure6-8 looks like that it was stretched vertically. Please make the river basin consistency in your figures. And please organize your figure6-8 better.*

We will modify Figures 6-8 accordingly and rearrange the subplots in them.

*8) Line 203, what is "a climatology"? please clarify it in details.*

With "climatology", we meant the mean ± one standard deviation over the study period (see caption of Figure 3). We will specify it also in the text in Section 3.1.

References:

Avanzi, F., Rungee, J., Maurer, T., Bales, R., Ma, Q., Glaser, S., and Conklin, M.: Climate elasticity of evapotranspiration shifts the water 370 balance of Mediterranean climates during multi-year droughts, Hydrology and Earth System Sciences, 24, 4317–4337, 2020

Marchina, C., Natali, C., and Bianchini, G.: The Po River water isotopes during the drought condition of the year 2017, Water, 11, 150, 2019

Masante, D., Vogt, J., McCormick, N., Cammalleri, C., Magni, D., and de Jager, A.: Severe drought in Italy – July 2017,https://doi.org/https://edo.jrc.ec.europa.eu/documents/news/EDODroughtNews201707_Italy.pdf , 2017

Toreti, A., Bavera, D., Acosta Navarro, J., Cammalleri, C., de Jager, A., Di Ciollo, C., Hrast Essenfelder, A., Maetens, W., Magni, D., Masante, D., Mazzeschi, M., Niemeyer, S., and Spinoni, J.: Drought in Europe August 2022, https://doi.org/doi:10.2760/264241, 2022a.

Toreti, A., Bavera, D., Avanzi, F., Cammalleri, C., De Felice, M., de Jager, A., Di Ciollo, C., Gardella, M., Gabellani, S., Leoni, P., Maetens, W., Magni, D., G., M., Masante, D., Mazzeschi, M., McCormick, N., Naumann, G., Niemeyer, S., Rossi, L., Seguini, L., Spinoni, J., and van den Berg, M.: Drought in Europe April 2022, https://doi.org/doi:10.2760/40384, 2022b.

Yang, X., Tetzlaff, D., Soulsby, C., Smith, A., and Borchardt, D.: Catchment Functioning Under Prolonged Drought Stress: Tracer-Aided Ecohydrological Modeling in an Intensively Managed Agricultural Catchment, Water Resources Research, 57, e2020WR029 094, 2021.

---

## Author Comment (AC2)

Dear Referee #2,

We would like to thank you for your review of our paper and your fruitful comments on it. Please find below our replies, in plain text, to your comments, in italic, with intended changes to the manuscript.

*The manuscript "Parameter transferability of a distributed hydrological model to droughts" evaluates a distributed hydrological model for the Po river basin during wet years and droughts of different severities. The authors have calibrated the model to multiple discharge stations and analyzed how well the model simulates drought conditions with respect to multiple variables (discharge, evaporation and total water storage). While this is an interesting and relevant topic, I recommend addressing the following major comment to make it a novel/unique publication:*

1. *As clearly stated in the introduction, previous studies have already illustrated hydrological models tend to poorly represent droughts. In its current version, the manuscript mainly illustrates that this is also the case for the hydrological model and study region used in this study. That is why I recommend bringing the study a step further. For example, would it be possible to really pin-point what exactly is causing this mis-representation during droughts? This could be done for example through a more extensive data analysis. This would allow gaining a better understanding of why the Po region is poorly modelled during droughts which can be used in future studies to improve the model representation. Alternatively, one could consider comparing different model improvement scenario's, but I can imagine this may be out of the scope of this particular study.*

We appreciate your hint to improve our study and we will add further analyses on where and how the simulation of streamflow (Q) degraded during the severe drought, and the potential causes for this. Specifically, we will adjust Figure 4 by grouping the boxplots in panel d into calibration and evaluation sub-catchments, and we will add in the supplement similar figures for the components of the Kling Gupta Efficiency to show that the decrease in Q simulation during the severe drought was mainly driven by an overestimation of Q in evaluation sub-catchments. Potential causes for this could be an over/underestimation of precipitation/streamflow data used to force/evaluate the model, an overestimation of the contribution of Terrestrial Water Storage to streamflow (Q) generation, and an underestimation of evapotranspiration (ET) during the severe drought. We will show that the observed difference between annual precipitation (P) and outgoing fluxes was 69±234 mm in 2012 and 51±202 mm in 2022, as mean ± 1 standard deviation across the study sub-catchments, and thus the uncertainty in observed data we used to force and evaluate the model has not increased systematically during the severe drought compared to the moderate droughts. Furthermore, at the outlet section the model slightly overestimated TWS both at the beginning and the end of 2022 (simulated TWS = -65 mm vs observed TWS = -92 mm in September 2021, and simulated TWS = -100 mm vs observed TWS = -158 mm in August 2022, Figure 5), and thus it did not overestimate its contribution to Q. Given the increased uncertainty in monthly ET standardized anomalies we detected for 2022 and these pieces of evidence, we suspect that the underestimation of ET was the main cause for the decrease in Q performances, especially for human disturbed sub-catchments. We will revise extensively Sections 3.2, 3.3, and 4.1 to show and discuss these points. Furthermore, we will emphasize better throughout the manuscript specific knowledge gaps that we addressed in our work (i.e., model performances during moderate vs severe droughts, the investigation of ET and storage as possible predictors for decreases in Q simulation, and model transferability from moderate to severe droughts) and the novelty of our work (e.g., model capabilities in

representing moderate droughts, but degraded transferability from moderate to severe droughts, and identification of possible causes for this).

> 2. *In addition, the language needs to be improved throughout the entire manuscript (see list of textual suggestions below for some examples). Also, there are some details missing in the description of the methods and results (see major/minor comments for some examples).*

We will revise thoroughly the language throughout the paper, by welcoming all your textual suggestions and a further round of proof-reading with a particular focus on language.

*Major comments:*

> 3. *Section 2.4.1: This section needs to be restructured and formulated more concise as it is currently confusing. This also impacts the understanding of the results section. For example: I recommend mentioning first the calibration/validation setup (so which years do you use for calibration and which for validation) and after that those precipitation numbers (or better: put them in a table). That would improve the readability a lot. Also, you write that you use two periods for calibration and three for validation. Do you use all three validation periods for both calibration scenarios? You mention here two calibration periods, but in Section 3.2 you only present results for one of those calibration scenario's (this becomes clear after having read Section 3.4, but should be clear from the start). What about the remaining years in 2009 – 2022 (so 2009-2011, 2013, 2015, 2021)? It feels like a waste not to use those too.*

We will reconstruct Section 2.4.1 to make our approach easier to grasp. Specifically, we will state first the calibration and validation periods, then their annual P anomalies, and we will point to the Sections where we present the results for the two calibration experiments. Further, we will use our whole study period for a general evaluation of model performances to make the best out of the entire dataset we collected. We will present the results from this evaluation in Section 3.2 for the model calibrated during "normal" years and in a table in the supplement for the other calibration experiment, along with evaluation scores for each calibration and evaluation period.

> 4. *Section 2.4.2: So each catchment was calibrated individually? How were parameters then transferred in space to the remaining regions? You mention here running the model for two years, while in line 98 you mention the entire 2009-2022 period. You mention a spin-up time of 6 months, but in line 98 you write 1 year. You use 18 of the 38 stations for calibration. Why those? How do you use the remaining stations? It feels like a waste not to use all the data available.*

We will add in Section 2.4.2 more details about the calibration procedure we used, fully described in Alfieri et al. (2022) we referred to. We calibrated four model parameters: three are spatially distributed and one is lumped for the whole domain. We set the first guess parameters from (i) maps of soil characteristics and land cover for the three distributed parameters, and (ii) authors' previous experience for the lumped parameter. Then we used an iterative parallel search algorithm to rescale the first guess parameters to minimize the cost function. This allowed us to preserve the spatial patterns from the first guess parameters, while minimizing the cost function. We will mention explicitly that we used a 1-year warm-up period for the simulation over the whole study period and 6-month warm-up periods for the calibration runs for computational reasons. Finally, we acknowledge that in the original manuscript we did not explain sufficiently the choice of the sub-catchments we used for calibration. We selected the same calibration

sub-catchments as Alfieri et al. (2022), who investigated the integration of remote sensing products into hydrological modelling over the same study area, in order to be consistent between these two studies. We will state this more clearly in Sections 2.3.1 and 2.4.2, as well as we will link our general evaluation of model performances to Alfieri et al. (2022) in Section 4.1.

5. *Section 3.1: The phrase "ET or Q anomalies" is confusing. You then mean that absolute monthly values are higher/lower for a specific year compared to the monthly mean. However, "ET anomalies" would mean that the presented/visualized values are relative to a long-term mean (e.g. similar to TWSA).*

Thank you for pointing this inconsistency out; we will avoid referring to anomalies in Section 3.1, since here we present the monthly values of P, Q, ET, and TWS during the analyzed droughts compared to their climatology (monthly mean ± 1 standard deviation). More in general, with "anomalies" we meant the standardized anomalies computed according to Equation 2 and so, by removing the monthly means and standard deviations (see line 179). To avoid confusion, we will substitute it with "monthly standardized anomalies" throughout the manuscript and we will state more clearly in Section 2.4.3 that we are referring to values relative to the monthly climatology and not to the long-term mean.

6. *Section 3.3: How did you estimate "deviations from seasonality"? The phrase TWSA anomalies (i.e., total water storage anomalies anomalies), sounds very odd. I recommend mentioning results related to TWSA first and then ET (instead of going back and forth). Did you validate ET with respect to station data too (dots in Fig. 6)? If yes, what data did you use?*

We used the monthly standardized anomalies (see previous comment) as a measure for the deviations from seasonality and with "TWSA anomalies" we meant the monthly standardized anomalies in TWSA. For the sake of clarity, we will change it to "monthly TWS standardized anomalies" in the whole paper. Further, we welcome your suggestion and we will present the results for TWS simulation first and then for ET simulation in Section 3.3, by adapting Figure 5 too. Regarding your comment on Figure 6, please note that the dots in it correspond to the outlets of the study sub-catchments. Here we are presenting the results of ET evaluation at catchment scale for all the study sub-catchments, complementary to the results for the outlet section (Figure 5). An evaluation of the remote sensing-based product against in-situ ET data across Italy and even during drought events is described in Bruno et al. (2022).

7. *The authors conclude the misrepresentation of ET is the reason for the poor Q performance during severe droughts. But why is ET poorly represented? Would you get the same result with a different satellite product? What processes could cause this? Also, when analyzing and describing results, they did not consider uncertainties in the precipitation, evaporation and total water storage observations (which could affect the results considerably) nor human actions (even though the Po river basin is heavily influenced by humans as stated by the authors in the manuscript).*

The misrepresentation of ET during the severe drought may be caused by irrigation, which is not included in the model and strongly increases water availability for ET during severely dry periods, and the uncertainties in model structure and parameterization for ET in water-limited conditions. We will add more discussion about it in Section 4.1. Moreover, we will explicitly mention in Section 4.2 that using different ET products to benchmark model simulations could be a possible way forward for future work. Regarding the uncertainty in the datasets we used to force and evaluate the model, we will present in Section 3.3 and discuss in Section 4.1 an estimation of their uncertainty to show that this has not increased

systematically during the severe drought and therefore, it seems implausible as main cause for the overestimation in Q we detected (see reply to comment 1). Finally, we considered possible human influences in the study area in the analysis by stratifying the evaluation of spatial patterns by land cover type and, more in general, we discussed them in Section 4.2.

*Minor comments:*

   8. *Line 2: Please be more specific: What feedback mechanisms for example?*

We are referring to mechanisms such as the climate elasticity of ET to P deficits sustained by carryover storage; we will rephrase it.

   9. *Line 8: One could argue KGE = 0.59 does not indicate the model is performing "well" (line 7).*

We will rephrase it.

   10. *Line 17: If I'm not mistaken, you are not giving any "guidelines" in this manuscript.*

We will rephrase it.

   11. *Line 20: Please be more specific: What "multifaceted impacts" are you referring to?*

We are referring to the impacts droughts can have on environment, human society, and economy; we will expand this point to be more specific.

   12. *Line 21: Please be more specific: Why/how does a warming climate lead to increased drought impacts?*

Naumann et al. (2021) shows an expected increase of drought impacts in a warming climate, driven by an increase in drought hazard (i.e., longer, more severe, and more frequent drought events); we will expand this point to be more specific.

   13. *Line 29: How do you define "streamflow droughts"?*

With streamflow droughts we meant deficits in streamflow, following Van Loon (2015); we will expand this point to be more specific.

   14. *Line 37: Why are the results inconclusive?*

Calibration during droughts showed increased model performances in the simulation of dry conditions compared to those achieved by calibrating during wet periods (Li et al., 2012; Yang et al., 2021), but still poor transferability to extreme drought conditions (Avanzi et al., 2020). Therefore, we argue that there is no general consensus in this regard. We will expand this point.

   15. *Line 54: It would be interesting to not only analyze whether model deterioration is related to ET or TWSA, but also why that is and what that tells us.*

We will reformulate the second research question accordingly.

   16. *Line 65: The sentence is a bit confusing. Percentage of what? Please move "of the whole country" closer to the first % value.*

We will rephrase it, thank you for the suggestion.

*17. Line 67: The Swiss region mentioned here is not shown in the figure referred to here.*

We will add Swiss boundaries in the overview map in panel d of Figure 1.

*18. Line 69: What does a.s.l. stand for?*

It stands for above sea level; we will expand it.

*19. Line 74: Isn't annual discharge (mm/year) automatically cumulative?*

We will correct it.

*20. Line 80: Please show the three major lakes more clearly in Fig. 1.*

We will modify panel c in Fig. 1 to make them clearer.

*21. Line 82: Please be more specific: How much (%) of the water use is related to irrigation?*

According to Authority Po River Basin (2006), 60% of surface water withdrawals in the river basin is used for irrigation; we will rephrase the paragraph by specifying it.

*22. Line 96: Model outputs can be several things, but I think here you mean "fluxes".*

We will change it.

*23. Line 107: Model inputs can be several things, but I think here you mean "forcing data".*

We will change it.

*24. Line 110: How accurate are these maps? What is the density of the field observations underlying these maps?*

The P product we applied as model forcing blends data from 1377 P gauges over the study area (Alfieri et al., 2022) with radar observations. This product was compared with other P products over the study region and it outperformed gauges-only interpolation (Bruno et al., 2021) and other satellite products for hydrological modelling (Alfieri et al., 2022). Therefore, we expect uncertainties from the P product to be as low as possible from state-of-the-art products, despite acknowledging that in specific regions of the study area, such as the mountainous ones, may still be relevant (Avanzi et al., 2021). We will add more more details about the P product in Section 3.2.1.

*25. Line 119: "around 5km" is a vague formulation*

We will rephrase it by providing the resolution of the remote sensing-based dataset at the sub-satellite point.

*26. Line 120: Which energy model specific?*

The surface energy model used by the LSASAF product to derive ET estimates from remote-sensed data is based on a Soil-Vegetation-Atmosphere Transfer scheme described in Ghilain et al. (2011) we referred to; we will specify it.

*27. Line 122: Please start new paragraph at "Finally, we employed…"*

We will do it.

*28. Line 134: Which technique did you use for the regridding?*

We used a nearest neighbour technique to regrid the datasets; will expand this point.

*29. Line 169: How exactly did you evaluate the spatial variability? Reading the results section, I understand you did that through visual comparison?*

We evaluated the model capabilities in reproducing ET and TWS spatial variability by computing pixel-wise deviations. We will rephrase Section 2.4.3 to specify it.

*30. Line 187: How would the results be affected if you did not normalize? In other words, is the normalization really needed?*

We normalized the Root Mean Square Error (RMSE) to the observed standard deviations to allow for a fair comparison among sub-catchments that may have different ranges in observations; however, we do not expect the normalization to affect substantially our results, since we are computing RMSE for monthly Q, ET and TWSA expressed in mm.

*31. Line 194: How do you define "duration and severity of P deficits"? When reading this, I think of drought duration & severity; see for example Section 3.3 in the following paper: S. Huang, Q. Huang, J. Chang, G. Leng. Linkages between hydrological drought, climate indices and human activities: a case study in the Columbia river basin. Int. J. Climatol., 36 (1) (2016), pp. 280-290, https://doi.org/10.1002/joc.4344. However, I don't think this is what you mean. You did calculate rainfall anomalies which are the numbers you are referring to here. So please be clear with what you mean.*

Yes, we used the standardized anomalies as a measure of P deficits. To make it clearer, we will add more details on how we are identifying and characterizing drought years in Section 2.4.1.

*32. Line 204: Avoid using the term "historical minimum" since your study period is pretty short.*

We will rephrase it.

*33. Section 3.2: Please mention at the beginning that all results are validation results when calibrating with respect to the years 2018/19.*

We will welcome the suggestion.

*34. Line 219: Are you sure you mean Fig. 1?*

We thank you for pointing this misreference out; we will amend it.

*35. Line 229: Why did the model have difficulties in reproducing ET deviations from the seasonality? What could explain this?*

Please see our reply to comment 5.

*36. Line 252: Please show the validation results too.*

Please note that in Figure 8 we show the validation results (lines 254-257).

*37. Line 273: Better than what?*

Among the models they analyzed; we will rephrase it.

> *38. Line 307: "we showed the value of remote sensing" Not quite, you merely used remote sensing data for your analysis.*

We will rephrase it.

> *39. Figures: Please ensure the grid spacing is the same in all figures. Currently, the Po river basin looks differently in Fig. 1,4 vs. Fig. 6-8.*

We will modify Figures 6-8 accordingly.

> *40. Textual comments:*
>
> *Line 3: "to gain more insights" -> "to gain more insight"*
>
> *Line 5: "climatic conditions of the calibration period" -> "climatic conditions during the calibration period"*
>
> *Line 8: remove second "=" sign*
>
> *Line 9: "The model simulated well Q for the outlet section of the basin" -> "The model simulated Q well at the Po basin outlet"*
>
> *Line 38: "model deterioration in Q simulation" -> "decreased Q performances"*
>
> *Line 41: "while" -> "but"*
>
> *Line 47: "so" -> "hence"*
>
> *Line 52: "To contribute to fill this research gap" -> "To contribute to filling this research gap"*
>
> *Line 52: "drop" -> "decrease"*
>
> *Line 57: "in northern Italy and the flood- and drought-rich period" -> "in northern Italy during the flood- and drought rich period"*
>
> *Line 58: "… and we evaluated the modelling capabilities in…" -> "and evaluated the model's capability in …"*
>
> *Line 59: " for the whole river basin and 38 sub-catchments" -> "for the whole river basin and its 38 sub-catchments"*
>
> *I'm stopping here with writing down textual suggestions.*

Thank you very much for your suggestions for improving the language. We will change the text accordingly and add a further round of proof-reading with a focus on language.

References:

Alfieri, L., Avanzi, F., Delogu, F., Gabellani, S., Bruno, G., Campo, L., Libertino, A., Massari, C., Tarpanelli, A., Rains, D., et al.: Highresolution satellite products improve hydrological modeling in northern Italy, Hydrology and Earth System Sciences, 26, 3921–3939, 365, 2022

Authority, P. R. B.: Caratteristiche del Bacino del Fiume Po e Primo Esame dell' Impatto Ambientale Delle Attivitá Umane Sulle Risorse Idriche (Characteristics of Po River Basin and First Analysis of the Impact of Human Activities on Water Resources), https://doi.org/adbpo.it/download/bacino_Po/AdbPo_Caratteristiche-bacino-Po_2006.pdf (in Italian), 2006.

Avanzi, F., Rungee, J., Maurer, T., Bales, R., Ma, Q., Glaser, S., and Conklin, M.: Climate elasticity of evapotranspiration shifts the water 370 balance of Mediterranean climates during multi-year droughts, Hydrology and Earth System Sciences, 24, 4317–4337, 2020

Avanzi, F., Ercolani, G., Gabellani, S., Cremonese, E., Pogliotti, P., Filippa, G., Morra di Cella, U., Ratto, S., Stevenin, H., Cauduro, M., and Juglair, S.: Learning about precipitation lapse rates from snow course data improves water balance modeling, Hydrol. Earth Syst. Sci., 25, 2109–2131, https://doi.org/10.5194/hess-25-2109-2021, 2021

Bruno, G., Pignone, F., Silvestro, F., Gabellani, S., Schiavi, F., Rebora, N., Giordano, P., and Falzacappa, M.: Performing Hydrological Monitoring at a National Scale by Exploiting Rain-Gauge and Radar Networks: The Italian Case, Atmosphere, 12, 771, 2021

Bruno, G., Avanzi, F., Gabellani, S., Ferraris, L., Cremonese, E., Galvagno, M., and Massari, C.: Disentangling the role of subsurface storage in the propagation of drought through the hydrological cycle, Advances in Water Resources, 169, 104 305, 2022

Ghilain, N., Arboleda, A., and Gellens-Meulenberghs, F.: Evapotranspiration modelling at large scale using near-real time MSG SEVIRI derived data, Hydrology and Earth System Sciences, 15, 771‑786, 2011.

Li, C., Zhang, L., Wang, H., Zhang, Y., Yu, F., and Yan, D.: The transferability of hydrological models under nonstationary climatic conditions, Hydrology and Earth System Sciences, 16, 1239–1254, 2012

Naumann, G., Cammalleri, C., Mentaschi, L., and Feyen, L.: Increased economic drought impacts in Europe with anthropogenic warming, Nature Climate Change, 11, 485–491, 2021

Van Loon, A. F.: Hydrological drought explained, Wiley Interdisciplinary Reviews: Water, 2, 359–392, 2015

Yang, X., Tetzlaff, D., Soulsby, C., Smith, A., and Borchardt, D.: Catchment Functioning Under Prolonged Drought Stress: Tracer-Aided Ecohydrological Modeling in an Intensively Managed Agricultural Catchment, Water Resources Research, 57, e2020WR029 094, 2021